# One-Pot Ionic Liquid-Mediated Bioprocess for Pretreatment and Enzymatic Hydrolysis of Rice Straw with Ionic Liquid-Tolerance Bacterial Cellulase

**DOI:** 10.3390/bioengineering9010017

**Published:** 2022-01-06

**Authors:** Malinee Sriariyanun, Nichaphat Kitiborwornkul, Prapakorn Tantayotai, Kittipong Rattanaporn, Pau-Loke Show

**Affiliations:** 1Biorefinery and Process Automation Engineering Center, Department of Chemical and Process Engineering, The Sirindhorn International Thai-German Graduate School of Engineering (TGGS), King Mongkut’s University of Technology North Bangkok, Bangkok 10800, Thailand; nichaphatkitii@gmail.com; 2Department of Microbiology, Faculty of Science, Srinakharinwirot University, Bangkok 10110, Thailand; prapakorn@g.swu.ac.th; 3Department of Biotechnology, Faculty of Agro-Industry, Kasetsart University, Bangkok 10900, Thailand; kittipong.r@ku.th; 4Department of Chemical and Environmental Engineering, Faculty of Engineering, University of Nottingham Malaysia, Semenyih 43500, Selangor, Malaysia; PauLoke.Show@nottingham.edu.my

**Keywords:** biorefinery, cellulase, enzymatic hydrolysis, ionic liquid pretreatment, lignocellulose biomass, one-pot process

## Abstract

Ionic liquid (IL) pretreatment of lignocellulose is an efficient method for the enhancement of enzymatic saccharification. However, the remaining residues of ILs deactivate cellulase, therefore making intensive biomass washing after pretreatment necessary. This study aimed to develop the one-pot process combining IL pretreatment and enzymatic saccharification by using low-toxic choline acetate ([Ch][OAc]) and IL-tolerant bacterial cellulases. Crude cellulases produced from saline soil inhabited *Bacillus* sp. CBD2 and *Brevibacillus* sp. CBD3 were tested under the influence of 0.5–2.0 M [Ch][OAc], which showed that their activities retained at more than 95%. However, [Ch][OAc] had toxicity to CBD2 and CBD3 cultures, in which only 32.85% and 12.88% were alive at 0.5 M [Ch][OAc]. Based on the specific enzyme activities, the sugar amounts produced from one-pot processes using 1 mg of CBD2 and CBD3 were higher than that of Celluclast 1.5 L by 2.0 and 4.5 times, respectively, suggesting their potential for further application in the biorefining process of value-added products.

## 1. Introduction

The biorefining process of lignocellulose biomass has been applied for the production of value-added products, e.g., biofuels, drop-in fuels, biochemicals, and platform chemicals, due to the abundance of raw materials and their compatibilities to the available current technologies and infrastructures [1]. In general, the biorefining process is composed of four main steps, including pretreatment, hydrolysis, fermentation/conversion, and recovery/separation. Among these steps, the pretreatment and hydrolysis steps constitute 4major portions of the total process cost (up to more than 50%) [2]. Pretreatment is a crucial step to enhance enzymatic hydrolysis efficiency through various mechanisms, such as removing lignin [3], altering cellulose crystallinity [4], increasing biomass porosity and contact area [5], and enhancing cellulose swollen and wettability [6]. Many pretreatment methods (including chemical, biological, mechanical, and physicochemical methods) have been demonstrated and characterized to understand their mechanisms to promote biomass hydrolysis, to reveal their limitations and to identify room for further improvements.

Ionic liquid (IL) is a salt that has a liquid state at a temperature lower than 100 °C. It is synthesized by mixing various types of anions and cations, either organic or inorganic ions [7]. IL is a green solvent with several interesting properties, such as low volatility, low viscosity, low corrosiveness, low melting point, high ionic conductivity, thermal stability, and recyclability [8]. Regarding to these properties, ILs have been applied in lignocellulose fractionation and pretreatment and demonstrated to have a high cellulose recovery rate and removal of lignin [9]. The effects of IL on lignocellulosic biomass have been proposed that IL’s anions interact with hydroxyl groups of cellulose, while IL’s cations act as the electron acceptors for electron pairing. These interactions disrupt the networks of intramolecular and intermolecular hydrogen bonds of cellulose fibrils and ultimately cause cellulose dissolution in IL [10]. Different combinations of cations and anions showed different efficiencies in cellulose dissolution, therefore, optimization and experimentation are necessary to select the suitable IL to maximize yields [11,12]. For example, 1-ethyl-3-methyl-imidazolium acetate ([EMIM][OAc]) is an IL that is often applied in lignocellulose pretreatment. [EMIM][OAc] pretreatment of sugarcane bagasse at a mild temperature (90 °C) increased glucose production from enzymatic hydrolysis to 95.2% and reduced the cellulose’s degree of polymerization [13]. Additionally, it also has good potential for the recycling process to reuse IL. [EMIM][OAc] used for lignocellulose pretreatments were demonstrated to be recycled for 3–5 rounds, and it still maintained the sugar recovery of more than 90% compared to the first round [14,15].

Choline acetate ([Ch][OAc]) is another IL that has been applied for lignocellulose pretreatment. Similar to [EMIM][OAc], [Ch][OAc] pretreatment on bagasse powder obtained a saccharification efficiency of up to 95% with a loading ratio of IL to biomass of 1.5, and the sugars produced from this study were converted to ethanol with 85% of the theoretical value [16]. Interestingly, [Ch][OAc] was demonstrated to be a relatively low toxic chemical to living organisms and less inhibitory to cellulase functions than other ILs [17]. For example, the presence of [EMIM][OAc] (at 2.0 M) in cellulase hydrolysis reaction reduced the sugar yields by more than 40% [18]. Therefore, intensive biomass washing after pretreatment before the hydrolysis step is crucial to ensure the full efficiency of cellulase function, and this washing leads to an increase in wastewater treatment cost, complexity in neutralization, and difficulty in the solid–liquid separation process. Different strategies were proposed to lessen IL toxicity, such as using low toxic IL, high tolerant enzyme and microorganism, and diluted IL [19].

One-pot process is another concept that can be applied by combining the pretreatment process and hydrolysis in one reactor to reduce the intensive washing step. Previously, two schemes of processes to produce ethanol from lignocellulosic biomass, (i) water washing (WW) between pretreatment and hydrolysis and (ii) one-pot, were comparatively analyzed in a techno-economic analysis [20]. With high biomass loading, the minimal ethanol selling prices (MESP) of the one-pot process and WW processes are 2.8 and 3.2 USD (14.29% difference in cost), respectively. The benefit of the one-pot process was related to the better cost reduction in enzyme and IL uses as it can offer recyclable process and it also helps to reduce the cost of wastewater treatment [20]. In this study, a one-pot process for the production of sugars from rice straw was developed by using a low toxic IL, [Ch][OAc], and IL-tolerant cellulases. The [Ch][OAc] pretreatment of rice straw was optimized to maximize the production of sugars from rice straw. The pretreated rice straw samples were further analyzed for their compositions and modifications in their functional chemicals to understand the mechanisms of how [Ch][OAc] pretreatment enhances enzymatic saccharification. The IL-tolerant cellulases were prepared from salted soil-inhabited bacteria and tested for their tolerances in [Ch][OAc] and [EMIM][OAc], and benchmarked with the commercial cellulase enzyme, Celluclast 1.5 L. Then, the efficiencies of IL-tolerant cellulases, in terms of sugar productions and specific enzyme activities, in a one-pot process were assessed without a water washing step. The results obtained from this study could contribute to the further development of the biorefining processing of lignocellulose for the production of value-added products.

## 2. Materials and Methods

### 2.1. Chemicals and Biomass Preparations

Rice straw biomass was collected from a local field in the central part of Thailand. The bulk biomass was transferred to the lab and dried with hot-air oven (Model. LLD-OF-155, LabLeader, Bangkok, Thailand) at 60 °C for 24 h until a constant dried weight was achieved. Rice straw was cut to reduce the size by blender and the particle size was screened by 20-mesh-sized aluminium sieve. Then, the biomass sample was kept in a sealed bag and placed at room temperature until use. The ILs used in this study are cholin acetate ([Ch][OAc]) and 1-ethyl-3-methyl-imidazolium chloride ([EMIM][OAc]) supplied by Sigma Aldrich (St. Louis, MO, USA). Commercial cellulase enzyme, Celluclast 1.5 L (produced by *Trichoderma reesei* ATCC 26921) and dinitrosalicylic acid (DNS) was purchased from Sigma Aldrich (St. Louis, MO, USA). The chemicals used for bacterial culturing were supplied by Himedia Laboratory (Mumbai, India) and other chemicals used in this study were purchased from Ajax Finechem (Brookline, MA, USA).

### 2.2. Lignocellulose Pretreatment and Enzymatic Saccharification

The [Ch][OAc] pretreatment in this study was optimized based on the Response Surface Method (RSM) to maximize the productions of sugars from rice straw. The Box-Behnken design (BBD) was employed to RSM experiment by using Design-Expert Software (version 7.0.0 Stat-Ease Inc., Minneapolis, MN, USA) as a previous study [4]. The three pretreatment parameters, including pretreatment temperature (X_1_: 90–130 °C), pretreatment time (X_2_: 60–360 min) and biomass loading ratio (X_3_: 5–15% wt), were varied to three levels of testing. The amounts of reducing sugars were analyzed by using DNS assay [21] and considered to be a response factor (Y). A total of 17 experimental runs with different testing levels of each pretreatment parameter and the amounts of reducing sugars (Table 1) were analyzed based on one-way analysis of variance (ANOVA). The effects of each pretreatment parameter and the interaction of pretreatment parameters on the sugar yields were determined as the regression coefficients and visualized in the contour maps.

Each [Ch][OAc] pretreatment was conducted in a screw-capped glass tube containing 1 g of rice straw sample and [Ch][OAc] in the biomass loading ratio of 5 wt.%, 10 wt.%, and 15 wt.%. Subsequently, each tube was placed in the controlled-temperature hot-air oven to the targeted pretreatment temperature at 90, 110, and 130 °C for 60, 210, and 360 min depending on the designed run (Table 1). After pretreatment, the pretreated biomass was recovered by the addition of anti-solvent, deionized water with 1:1 (*w*/*w*) ratio to IL. The sample was centrifuged (Model. Combi 514R, Hanil Scientific Inc, Gimpo, Korea) at 8000× *g* for 10 min to separate solid and liquid fractions. The solid biomass fraction was washed with deionized water 5 times with 1:1 (*w*/*w*) ratio to IL to remove [Ch][OAc] residues. The washed samples were dried in hot-air oven at 60 °C for 24 h until a stable weight was achieved. The pretreated samples were subsequently hydrolyzed in enzymatic saccharification.

The enzymatic saccharification was conducted in a screw-capped glass tube. Each reaction contained 0.5 g of rice straw, 20 mL of 50 mM sodium acetate buffer (pH 5.0), 200 μL of 2 M sodium azide, and 20 FPU/g-biomass of Celluclast 1.5 L (Sigma Aldrich, St. Louis, MO, USA, produced by *Trichoderma reesei* ATCC 26921) [12]. The mixture was incubated in a rotary shaker set at 50 °C and 200 rpm for 72 h. The reaction was terminated by heating in a water bath (Model. DWB20-P, DLAB Scientific Co., Ltd., Beijing, China) at 100 °C for 10 min, and the liquid hydrolysate was collected by centrifugation at 8000× *g* for 10 min [12]. The concentrations of reducing sugars in liquid hydrolysates were determined by using 3,5-dinitrosalisylic acid (DNS) assay [21]. The compositions of cellulose, hemicellulose and lignin of untreated and pretreated samples were analyzed based on Van Soest method [22]. Each sample was repeated three time to confirm its validity.

### 2.3. FT-IR Spectroscopy Analysis

The chemical modifications on pretreated lignocellulose samples were analyzed by using Fourier transform infrared (FT-IR) spectrometry (Spectrum 2000, Perkin Elmer, Waltham, MA, USA). The FTIR spectra of untreated and pretreated rice straw samples were recorded from the wavenumber interval of 1800–800 cm^−1^. The spectra of both samples were compared to indicate the changes in contents and chemical arrangements of cellulose, hemicellulose, and lignin.

### 2.4. Preparation of IL-Tolerant Cellulase

In this study, two cellulase-producing bacterial strains, CBD2 and CBD3, were cultured for cellulase preparation. CBD2 and CBD3 were isolated from saline soil samples collected from a local shrimp farm in the central part of Thailand. The ability for cellulase production was screened by plating assay by spotting the culture on the carboxymethylcellulose (CMC) agar plate (containing 1% CMC, 0.2% NaNO_3_, 0.1% KH₂PO₄, 0.05% MgSO₄·7H₂O, 0.05% KCl, 0.2% Peptone) [23] and incubated for 24 h at 45 °C. The tested cultures were stained with 1% *w*/*v* congo red solution and washed with 1 M NaCl. The diameter of appeared clear halo was measured and compared with *Cellulomonas* sp., a cellulase-producing bacteria (provided by Thailand Institute of Scientific and Technological Research (TISTR)). After screening for cellulase-producing ability, CBD2 and CBD3 were identified by nucleotide sequencing for their 16s rDNA fragments with universal primers (27F and 1429 R) as described in our previous study [23]. The sequences of 16s rDNA fragments of CBD2 and CBD3 were aligned into the nucleotide BLAST tool via the National Center for Biotechnology Information (NCBI database) to identify the possible genera of these bacterial isolates. The nucleotide sequences of 16s rDNA fragments of CBD2 and CBD3 were deposited in the GenBank with accession No. OL687449 and OL687452, respectively.

After genera identification, CBD2 and CBD3 bacteria were challenged with NaCl in 1 L of CMC culture media (containing 1% CMC, 0.2% NaNO_3_, 0.1% KH₂PO₄, 0.05% MgSO₄·7H₂O, 0.05% KCl, 0.2% Peptone, and 1 M NaCl) and incubated for 48 h at 45 °C with shaking speed at 200 rpm. The cellulase was prepared from bacterial cultures based on the protocol of our previous work [23]. The supernatant fractions containing cellulase enzymes were collected by centrifugation at 8000× *g* for 15 min. The bacterial cellulase enzyme was further concentrated by the addition of ammonium sulphate precipitation to 80% saturation. The enzyme precipitate was formed during incubation at 4 °C for 24 h, and it was harvested by centrifugation at 12,000× *g* in a refrigerated centrifuge for 15 min. The enzyme precipitate was redissolved in 10 mL of 50 mM sodium phosphate buffer (pH 7.0) and desalted by using Float-A-Lyzer dialysis membrane with 10 kDa MWCO (Sigma Aldrich, St. Louis, MO, USA). The desalting was conducted by changing the phosphate buffer three times every 2 h. The dialyzed enzyme was concentrated by using Vivaspin-500 column with 10 kDa MWCO (Sartorius, Göttingen, Germany) [23]. This concentrated enzyme was called as a crude enzyme in this study.

To characterize the molecular size of prepared cellulase, the crude cellulase was fractionated by using size-exclusion liquid chromatography (16 mm × 600 mm; HiPrep 16/60 Sephacryl S-100 HR, GE Healthcare Life Science, Chicago, IL, USA). The pre-packed column was equilibrated with 50 mM sodium phosphate buffer (pH 7.0) with a flow rate of 0.2 mL/min. Each eluate fraction was monitored for protein concentration by using Biorad protein assay (Bio-Rad Laboratory, Hercules, CA, USA) with spectrophotometry at wavelength 595 nm. Fractions with high enzyme activity were pooled and concentrated again by the Vivaspin-500 column for SDS-PAGE and zymogram analysis [23].

### 2.5. SDS-PAGE and Zymogram Analysis

The cellulase fraction with high cellulase activity obtained from size-exclusion liquid chromatography was mixed with loading buffer (containing 100 mM Tris-HCl (pH 6.8), 4% *w*/*v* SDS, 0.2% *w*/*v* bromophenol blue, 20% *v*/*v* glycerol, 200 mM DTT) at 1:1 ratio of volume and heated to full denaturation in boiling water for 5 min. The enzyme sample was loaded into precasted 12% SDS-PAGE gel (containing 4 mL of 30% Acrylamide/Bisacrylamide solution, 2.5 mL of 1.5 M Tris pH 8.8, 0.1 mL of 10% SDS, 0.1 mL of 10% ammonium persulfate, 4 µL of TEMED) and separated under the electrical field of electrophoresis at 100 V [23]. The cellulase protein was visualized by staining with Coomassie Blue R-250 solution (containing 0.25 g/L Coomassie Blue R-250, 500 mL/L methanol, 100 mL/L glacial acetic acid, and 400 mL/L distilled H_2_O) [23] and the molecular size was estimated by compared with protein standard marker (BLUeye Prestained protein ladder, GeneDireX, Tuoyuon, Taiwan).

For zymogram analysis, as in our previous study [23], the cellulase sample was mixed with loading buffer without heating and loaded into 12% gel SDS-PAGE (containing 1% CMC for zymogram analysis). After electrophoresis, the gel was soaked in 30 mM sodium phosphate buffer and 40% isopropanol (pH 7.2) for 1 h to remove SDS residue from the gel and followed with soaking in equilibrated buffer (containing 30 mM sodium phosphate buffer (pH 7.2)) for 1 h. To renaturate cellulase, the gel was soaked in 30 mM sodium phosphate buffer, 5 mM β-mercaptoethanol and 1 mM EDTA (pH 7.2) at 4 °C overnight. The gel was then stained in 1% Congo red solution for 15 min at room temperature and washed with 1 M NaCl until the clear zone against the red background was observed.

### 2.6. Cellulase Enzyme Assay

Endo-β-1,4-glucanase activity of cellulase was measured based on protocol of our previous study [23] by mixing 5% (*v*/*v*) of crude cellulase enzyme with 1% (*w*/*v*) CMC substrate in 30 mM sodium phosphate buffer (pH 7.0) in a 15 mL screw-capped glass tube and the sample was placed in a shaking incubator at 45 °C for 60 min with shaking speed 200 rpm. The enzyme was deactivated by heating at 100 °C for 5 min, and the liquid hydrolysate was separated from the mixture by centrifugation at 10,000× *g* for 10 min. The reducing sugar concentration in the liquid hydrolysate was quantitated by dinitrosalicylic acid (DNS) assay [21] by comparing it with the standard curve of glucose. Cellulase unit activity was calculated and defined as the amount of enzyme that releases 1 μmol of glucose per min at the tested condition. The enzyme specific activities were defined as the unit activity per milligram of the enzyme at the tested condition [23].

To find the optimal condition, in terms of pH and temperature of bacterial enzymes, the crude enzyme activities were measured under different pH and temperature. The first set of experiments were conducted in pH range of 3–10 in different types of buffers (pH 3.0–6.0 in sodium acetate buffer, pH 7.0–8.0 in sodium phosphate buffer, and pH 9.0–10.0 in Tris-HCl buffer) [23]. Each reaction was incubated in a shaking incubator at 50 °C for 60 min, 200 rpm. Then, the enzyme activity was stopped by heating at 100 °C for 5 min, and the liquid hydrolysate was separated from a mixture by centrifugation at 10,000× *g* for 10 min. The second set was conducted in a temperature range of 40–70 °C in sodium phosphate buffer (pH 7.0). Each reaction was incubated in a shaking incubator at 50 °C for 60 min, 200 rpm, and the enzyme was deactivated. The reducing sugar concentration was analyzed by DNS assay.

To evaluate the IL tolerance of cellulase, two types of ILs [Ch][OAc] and [EMIM][OAc] at 0.5, 1.0, and 2.0 M were added in the CBD2, CBD3. And Celluclast 1.5 L enzymes. Based on the protocol of our previous work [12], the cellulase, IL and 150 mM CMC substrate were mixed in 30 mM sodium phosphate buffer (pH 7.0) and placed in a shaker incubator at 45 °C, 200 rpm, for 1 h. The remaining enzyme activities were calculated based on the activity of control set without additions of Ils.

### 2.7. IL Toxicity Test to Bacteria Cells

To test the toxicity of Ils on the survival of CBD2 and CBD3 cultures, the cultures of these bacteria were grown in CMC broth media for 24 h at 45 °C, 200 rpm. The bacterial cells were collected with low-speed centrifugation at 3000 rpm for 10 min and the 10^8^ bacterial cells were resuspended in fresh 1 mL CMC media containing 0.5, 1.0, and 2.0 M of [Ch][OAc] and [EMIM][OAc] for testing and without the addition of IL for the control set. These testing cultures were incubated in shaking incubators for 16 h at 40 °C, 200 rpm. Then, the numbers of survived cells in IL testing cultures were counted by using the plating technique of serial dilution and spreading on a CMC agar plate [23]. The colony numbers of each dilution grown on the CMC agar plates were counted as colony-forming units (CFU) after 16 h.

### 2.8. One-Pot Process Set Up

The one-pot process of this study was newly set up to conduct the [Ch][OAc] pretreatment and enzymatic hydrolysis in one pot without washing step in between. A total of 0.5 g of rice straw biomass was pretreated with the optimal pretreatment condition obtained from the RSM study (pretreatment temperature, 129.21 °C; pretreatment time, 331.82 min; loading ratio, 10.68%). Then, the pretreated biomass in [Ch][OAc] was diluted to obtain the concentrations of [Ch][OAc] at 0.5, 1.0, and 2.0 M by addition of phosphate buffer (pH 5.0) (containing 200 μL of cellulase, 200 μL of 2 M sodium azide). For the control set, 0 M, the pretreated sample was water-washed as did in the RSM experiment to remove IL residues. The mixture was placed in a rotary shaker set at 50 °C and 200 rpm for 72 h. After termination of enzyme activities, the liquid hydrolysate was collected by centrifugation at 8000× *g* for 10 min. The concentrations of reducing sugars in liquid hydrolysates were determined by using DNS assay and the specific enzyme activities were calculated based on the given definition [21].

### 2.9. Statistical Analysis

The experimental results were analyzed for significance statistically using SPSS version 26.0. The difference in activities of each bacterial cellulases were analyzed by one-way ANOVA with a significance level of *p* < 0.05.

## 3. Results and Discussion

### 3.1. Optimization of [Ch][OAc] Pretreatment

In this study, RSM was used for the optimization of the [Ch][OAc] pretreatment to improve the enzymatic saccharification by cellulase enzyme for the production of sugars. The three pretreatment parameters that were varied include pretreatment temperature (X_1_), pretreatment time (X_2_), and biomass loading ratio (X_3_) (Table 1), as these parameters were easily adjustable and were demonstrated in previous works to influence pretreatment efficiency [3,4]. The pretreatment efficiency was determined based on the sugar yield (Y) released from the hydrolysate of pretreated rice straw measured using a DNS assay (Table 1). The results obtained from the RSM were used for the analysis of individual and interactive effects of pretreatment parameters on the sugar yield and the determination of optimal pretreatment condition to maximize sugar yields. The sugar yields obtained from 17 runs of the RSM design ranged from 120.47 mg/g in run No.1 to 528.30 mg/g in run No.17, suggesting the importance of optimization (Table 1).

The influences of pretreatment parameters on sugar yields were statistically analyzed by ANOVA and reported as an F value (Table 2). The linear vs. mean model was suggested to represent the experimental data of [Ch][OAc] pretreatment with the coefficient of determination (R^2^) value of 0.9549. The lack of fit analysis obtained a *p*-value of 0.3095, which suggested that the model was suitable to fit with the experimental data. The “Prob > F” value of the model was less than 0.0001, implying that the model was significant. The model F value of 146.21 implies the model is significant. There is only a 0.01% chance that a “model F value” this large could occur due to noise. The model terms, pretreatment temperature and pretreatment time, also had *p*-values less than 0.0001, suggesting the significant effects of these two parameters on sugar yields (Table 2). As a result, the mathematical model representing the effects of pretreatment parameters on sugar yields was generated and it was used to predict the optimal pretreatment condition to maximize sugar yields, as shown in Table 3. The predicted sugar yield obtained from the optimal pretreatment, at a pretreatment temperature of 129.21 °C, pretreatment time of 331.82 min, and a loading ratio of 10.68%, was 542.3 mg/g-biomass.

To validate the generated mathematical model of [Ch][OAc] pretreatment, a rice straw sample was pretreated with [Ch][OAc] by using the predicted optimal condition, and sugar yield was measured after enzymatic hydrolysis. The experimental sugar yield was 571.2 mg/g-biomass, which agreed well with the predicted yield with an error rate of only 1.64%. Compared to the untreated biomass, the sugars yields were increased by 6.09-fold (from 90.5 to 551.2 mg/g-biomass), suggesting the significance of the [Ch][OAc] pretreatment of rice straw (Table 3). In a previous study, pretreatments of rice straw with organic acid (citric acid, oxalic acid, and acetic acid) and inorganic acid (HCl) produced the highest sugar yield at 416.8 mg/g-biomass [4]. Thus, the results in this study supported the higher efficiency of IL pretreatment over traditional chemicals [24]. Furthermore, the response surface plot was generated to visualize the effects of two or more pretreatment parameters on sugar yields at a time (Figure 1). It could be observed from the trends in the response surface plot that both pretreatment temperature and pretreatment time had a direct positive effect on sugar yields. The higher the temperature of the pretreatment condition, the higher the sugar yields. Likewise, the longer the pretreatment time, the more sugars that were released from biomass. The synergistic effect of these two pretreatment parameters is observed at the red-colored zone of the response surface plot that represents the highest yields of sugars (Figure 1).

### 3.2. Modifications of Lignocellulose Biomass by [Ch][OAc] Pretreatment

To understand the effects of [Ch][OAc] pretreatment on rice straw biomass, its compositions, cellulose, hemicellulose, and lignin, before and after [Ch][OAc] pretreatments were analyzed (Table 3). After pretreatment, it was observed that the cellulose content was enriched from 40.52% to 45.84% and hemicellulose content increased from 32.70 to 34.03%, whereas the lignin content was reduced after pretreatment from 14.95 to 8.39% (Table 3). IL pretreatment was demonstrated previously to be an excellent solvent for removing lignin from lignocellulose biomass [25]. Therefore, the cellulose and hemicellulose contents were enriched after lignin removal from the biomass. Additionally, IL pretreatment reduced the crystallinity index of cellulose and interfered with the arrangements of chemical bonds between the lignocellulose composition to allow better accessibility of cellulase enzymes to biomass, thus enhancing the enzymatic saccharification [26].

FT-IR spectroscopy was conducted to analyze pretreated and untreated biomass to decipher the modifications of chemical compositions of rice straw caused by [Ch][OAc] pretreatment (Figure 2). It could be observed that most of the FT-IR spectra of both pretreated and untreated samples were similar, but some peaks were different in the intensities of transmittance and were related to the chemical bonds of cellulose, hemicellulose, and lignin. The heights of spectral peaks at 1515 cm^−1^ (representing aromatic skeletal vibrations of lignin [27]) and 1604 cm^−1^ (corresponding to stretching of aromatic benzene ring in lignin) were reduced in pretreated biomass compared to the untreated sample, indicating the removal of lignin content [28]. The transmittances of peaks at 1249 cm^−1^ and 1732 cm^−1^ corresponding to the C–O stretching and carbonyl C–O stretching linkages and ester linkages between lignin and hemicellulose were reduced in pretreated rice straw, suggesting the dissociation of hemicellulose and lignin networks by pretreatment [29]. This disintegration of hemicellulose and lignin networks could promote the enzymatic hydrolysis of biomass as it helps to break down the lignocellulose fibers. In addition to lignin removal and disintegration of hemicellulose and lignin networks, the heights of two peaks at 1105 cm^−1^ (corresponding to the C-O bonds of cellulose and hemicellulose [30]) and 1430 cm^−1^ (associated to the bending vibration of CH_2_ group [31]) were reduced in the pretreated sample compared to untreated sample. The intensities of two peaks were strong in crystalline cellulose, suggesting that the [Ch][OAc] pretreatment alters the crystalline to amorphous cellulose structure (Li et al. 2009) and promotes the enzymatic hydrolysis in the next step. The FT-IR analysis used to monitor the change of chemical composition and functional groups correlated well to the composition analysis of lignocellulose in Table 3. Therefore, based on the improved sugar yields and modifications in chemical compositions, the importance of [Ch][OAc] pretreatment was demonstrated for the promotion of the production of sugars from lignocellulose biomass.

### 3.3. Characterization of Cellulases Produced by Bacterial Strains Isolated from Saline Soil Samples

To develop the one-pot biorefining process so that it was compatible with [Ch][OAc] pretreatment, two bacterial strains, CBD2 and CBD3, isolated from saline soil samples collected from a local shrimp farm, were screen tested for cellulase-producing efficiency by using congo-red plating assay and compared with *Cellulomonas* sp. bacterium. Several studies reported the correlation of salt-tolerant microorganisms to IL tolerance [32,33,34]. For example, cellulase produced from *Paenibacillus tarimensis* that was isolated from the Tunisian salt lakes still retained its activity for 90% in 20% [EMIM][OAc] [35]. This correlation implies that the responses of microorganisms to stress caused by salt and IL are similar. After screening, these two bacteria strains with potential as cellulase producers were identified based on the sequencing of 16s rDNA fragments. The alignment results of the 16s rDNA fragments of CBD2 and CBD 3 were similar to *Bacillus* sp. (Accession MN416319.1) and *Brevibacillus* sp. (Accession MT292327.1) at 92% and 99%, respectively. Therefore, the sequences of 16s rDNA fragments obtained from this study were deposited in the GenBank with accession No. OL687449 and OL687452, respectively.

To characterize the properties of cellulases produced from CBD2 and CBD3, these two bacteria were cultured in liquid culture media containing CMC as the main carbon source. After 48 h of culturing, extracellular fractions of bacterial cultures were collected and concentrated by ammonium precipitation and named the crude enzyme fraction in this study. The crude enzyme was fractionated to a total of 30 fractions by using size-exclusion liquid chromatography (Figure 3). Each collected fraction was tested for CMCase activities that were measured based on the amounts of reducing sugars. The highest CMC activities of CBD2 and CBD3 crude enzymes were presented in fractions No. 19 and No. 22, respectively (Figure 3). To determine the molecular weight of the CMCases of CBD2 and CBD3, the active fractions were analyzed by SDS-PAGE analysis and zymogram analysis. The results of SDS-PAGE analysis, after staining with Coomassie Blue R-250, showed the major bands of proteins of CBD2 and CBD3 at 60 and 30 kDa, respectively. The lighter red zones, after staining with congo red, were also observed in zymogram analysis, indicating the active CMCase at 60 and 30 kDa, respectively. The molecular weight sizes of the CMCases produced from CBD2 were in the same ranges as observed in other CMCases produced by *Bacillus* sp. (~29–97 kDa [23,36]). However, the size of CMCase produced from CBD3 was smaller than those reported in identified *Brevibacillus* sp. (~70–90 kDa [37,38]).

To find the optimal working condition of these CMCases, crude cellulases produced from CBD2 and CBD3 were tested with hydrolysis reactions using CMC as substrates at different ranges of pH and temperatures (Figure 4). For pH test, the hydrolysis buffer was prepared to a pH in the range of 3 to 10, and the reaction temperature was set at 50 °C. The optimal pH for the cellulases of CBD2 and CBD3 was at 7 and 5, respectively (Figure 4). On the other hand, to find the optimal temperature, the pH of the hydrolysis buffer was set at pH 7, when the temperatures of the reaction were ranged between 40–70 °C. The optimal temperatures of these cellulases were at 55 °C and 50 °C for CBD2 and CBD3, respectively (Figure 4). It was noted that both enzymes have broad a range tolerance to pH changes because the enzyme activities still retained more than 80% for the highest activities at the extremely acidic and alkaline conditions. However, the temperature tolerance of CBD2 and CBD3 were different. For CBD2, when the temperature was higher than 55 °C, the enzyme activity was clearly reduced and completely deactivated at 70 °C. In the case of CBD3, more than 80% of cellulase activity was retained in the range of 45–65 °C, suggesting that CBD3 cellulase had high temperature tolerance (Figure 4). After the determination of the optimal condition, the CMCase activities of CBD2 and CBD3 were calculated to be 0.62 and 0.73 U/mL, respectively (Table 4).

### 3.4. Evaluation of IL-Tolerance Properties of Bacterial Cellulases

Previously, the correlations of salt tolerance and ionic liquid tolerance in the bacterial special and bacterial consortium were reported [32,34]. Therefore, the crude cellulase enzymes produced from CBD2 and CBD3 were tested for their tolerance to ILs. In this study, two types of ILs, [Ch][OAc] and [EMIM][OAc], were added into the hydrolysis reactions of CMC substrates to obtain the final concentrations of ILs at 0.5, 1.0, and 2.0 M. Both [Ch][OAc] and [EMIM][OAc] were demonstrated to be excellent solvents for lignocellulose pretreatments to improve enzymatic saccharification. Although with lower concentrations of ILs in the range of 25%, good efficiency in pretreatment was still observed [39]. Unfortunately, the presence of IL with a lower concentration of 10% in enzymatic hydrolysis reaction had a negative effect on cellulase function [12,40]. The tested concentrations of ILs in this work at 0.5–2.0 M were equivalent to 8.16–32.64% for [Ch][OAc] and 8.51–34.04% for [EMIM][OAc]. In the hydrolysis reactions, 150 mM of CMC was used as the substrate, then the crude cellulases of CBD2, CBD3, and Celluclast 1.5 L were mixed with [Ch][OAc] and [EMIM][OAc]. The reducing sugars obtained from the hydrolysis reactions were compared with the control reaction, without the addition of IL, and calculated as the remaining enzyme activities (Figure 5). Under the influence of [Ch][OAc], it was observed that the enzyme activities of CBD2 and CBD3 were maintained at over 95%, and the activities were enhanced when the [Ch][OAc] concentrations were at 1.0 and 2.0 M, suggesting the IL[Ch][OAc] tolerance properties of these enzymes. On the other hand, Celluclast 1.5 L activities were reduced to 81.16% at 2.0 M [Ch][OAc]. Interestingly, the crude cellulase activities of CBD2, CBD3, and Celluclast 1.5 L were reduced to 70.45%, 67.44%, and 60.54% at 2.0 M [EMIM][OAc], respectively (Figure 5). These observations suggested that [EMIM][OAc] was more toxic to cellulases compared to [Ch][OAc], and crude cellulases of CBD2 and CBD3 had higher tolerances to [Ch][OAc] and [EMIM][OAc] than Celluclast 1.5 L.

### 3.5. Evaluation of IL-Tolerance Properties of Bacterial Cultures

In addition to the evaluation of IL tolerance of the cellulase enzymes, the IL tolerance properties of CBD2 and CBD3 bacterial cultures were assessed under the influence of [Ch][OAc] at 0.5, 1.0, and 2.0 M. After the incubation of CBD2 and CBD3 bacterial cultures with [Ch][OAc] for 16 h, the number of survived cells was counted by using plating technique and compared with the control set (without the addition of [Ch][OAc]). The survival rates of CBD2 and CBD3 cultures under the influence of 0.5 M [Ch][OAc] were 32.86% and 12.88%, respectively (Figure 6). Similarly, at 2.0 M [Ch][OAc], the survival rates of CBD2 and CBD3 cultures were reduced to 24.00% and 11.04%, respectively (Figure 6). These results suggest that [Ch][OAc] was toxic to CBD2 and CBD3 cultures, although their cellulases were tolerant to [Ch][OAc]. Previously, [Ch][OAc] was reviewed to have lower toxicity toward living organisms than other types of ILs [21]. However, a similar study on [Ch][OAc] toxicity showed that after treatment with just 1.5% wt of [Ch][OAc], the hatchability of *Artemia salina* was reduced by more than 20% [41]. Furthermore, the EC50 value of [Ch][OAc] toxicity on *Vibrio fisheri* was shown to be at 8 mM, with a testing period of 5 min [42]. Therefore, this finding could be the guideline to the design of a one-pot process to select crude cellulase for hydrolysis with a combination of [Ch][OAc] pretreatment.

### 3.6. Efficiency of IL-Tolerance Cellulases in One-Pot Process

To set up the one-pot process, the rice straw sample was pretreated with [Ch][OAc] at the optimal pretreatment condition (pretreatment temperature, 129.21 °C; pretreatment time, 331.82 min; loading ratio, 10.68%). After pretreatment, the mixture of pretreated rice straw and [Ch][OAc] was diluted with hydrolysis buffer containing crude cellulases of CBD2, CBD3, and Celluclast 1.5 L without the step of biomass washing to 0.5, 1.0, and 2.0 M. For the control set (0 M), the pretreated sample was washed as did in the RSM study to remove IL residues. Using the same loading volumes of cellulases to maintain the same volumes of reactions, the reducing sugar concentrations obtained from the hydrolysis reactions without [Ch][OAc] pretreatment by using Celluclast 1.5 L were the highest (18.88 mg/mL) compared to those using CBD2 (1.27 mg/mL) and CBD3 (1.78 mg/mL) (Figure 7). However, when one-pot reactions were conducted at the remaining concentration of 0.5 M [Ch][OAc] in the hydrolysis step, the sugar concentrations produced by using Celluclast 1.5 L dropped to 6.32 mg/mL. Interestingly, when the concentrations of [Ch][OAc] increased to 1.0 and 2.0 M, the sugar yields produced by Celluclast 1.5 L hydrolysis increased to 10.89 and 12.45 mg/mL, respectively (Figure 7). This observation suggested that the presence of [Ch][OAc] during the hydrolysis reaction assisted in functions of Celluclast 1.5 L in the one-pot process, while it had a clear inhibitory effect on Celluclast 1.5 L in a separated process (with the water-washing step) as shown in Figure 5. In the case of Celluclast 1.5 L, the presence of [Ch][OAc] in a one-pot process can either inhibit or enhance the cellulase activities depending on the concentration of [Ch][OAc]. Such a jeopardy scenario was explained previously as that, at a low concentration of IL and inorganic salt, the lignocellulose biomass still maintains its toughness and cellulase is inefficient to function. While at a higher concentration of IL and inorganic salt, the biomass substrate was altered under the influence of IL and inorganic salt, which allows greater accessibility of cellulase to react with the substrate. The Critical Concentration of Substrate (CCS) values in balance with a concentration of IL (or salt) in a tested system can be determined by varying the biomass and IL (or salt) concentration until the jeopardy behavior of cellulase is observed [18,43]. On the other hand, the sugar yields obtained from the one-pot process using crude cellulases of CBD2 and CBD3 increased when the remaining concentrations of [Ch][OAc] were increased (Figure 7A). These trends of the CBD2 and CBD3 cellulases in the one-pot process were correlated to the [Ch][OAc] tolerance (Figure 4).

When the specific enzyme activities for the rice straw hydrolysis of CBD2, CBD3, and Celluclast 1.5 L were calculated based on the sugar concentrations obtained from the one-pot process with 2.0 M [Ch][OAc], the specific enzyme activities of CBD2 (0.32 U/mg) and CBD3 (0.72 U/mg) were higher than that of Celluclast 1.5 L (0.16 U/mg) (Figure 7). With calculations based on the specific enzyme activities, the sugars produced from one-pot processes using 1 mg of enzymes, CBD2, CBD3, and Celluclast 1.5 L, were 248.8, 559.9, and 124.4 mg, respectively. These obtained specific enzyme activities suggest that using the same loading amounts of CBD2 and CBD3 crude cellulases can produce more sugar yields from rice straw than that of Celluclast 1.5 L by 2.0 and 4.5 times, respectively (Figure 7). Additionally, it was observed that the higher the concentrations of [Ch][OAc], the higher the specific enzyme activities that were obtained in CBD2 and CBD3, suggesting their potential for the one-pot process.

## 4. Conclusions

IL pretreatment has been demonstrated to be an efficient chemical for pretreatment to enhance enzymatic saccharification, however, the presence of IL residue has an inhibitory effect on cellulase. Therefore, intensive biomass washing after pretreatment is necessary and produces a large amount of wastewater. The one-pot process combining [Ch][OAc] pretreatment with IL-tolerant cellulase was developed in this work to reduce the washing step and subsequently reducing the processing cost and time. The [Ch][OAc] pretreatment condition was optimized based on RSM to obtain the highest sugar conversion at 542.3 mg/g-biomass. The mechanism of [Ch][OAc] pretreatment to enhance the enzymatic saccharification of rice straw was explained based on the results of composition and FT-IR analysis, showing that pretreatment removed lignin, disintegrated the linkage between hemicellulose and lignin, and altered the cellulose molecular arrangement from the crystalline to amorphous structure. The IL-tolerant cellulases in this study were produced from *Bacillus* sp. CBD2 and *Brevibacillus* sp. CBD3 that were isolated from saline soil in a shrimp farm. The IL-tolerance of cellulases was tested with two types of ILs, [Ch][OAc] and [EMIM][OAc]. It was found that the CMCase activities of CBD2 and CBD3 were retained at more than 95% in 0.5–2.0 M [Ch][OAc], whereas their activities were reduced to 70% in 2.0 M [EMIM][OAc], suggesting a lower inhibitory effect of [Ch][OAc] to cellulases. However, a low concentration of [Ch][OAc], at 0.5 M, showed toxicity to the CBD2 and CBD3 cells and reduced the survival rates of CBD2 and CBD3 to 32.86% and 12.88%, respectively. Therefore, the crude cellulases of CBD2 and CBD3 were applied in a one-pot process with [Ch][OAc] pretreatment. Calculated based on the specific enzyme activity, the sugars produced from one-pot processes using 1 mg of CBD2, CBD3, and Celluclast 1.5 L were 248.8, 559.9, and 124.4 mg, respectively. Altogether, this one-pot process with IL-tolerant cellulase has benefits to enhancing sugar production, to skipping the biomass washing step, and to avoiding solid–liquid separation between pretreatment and the hydrolysis step.

## Figures and Tables

**Figure 1 bioengineering-09-00017-f001:**
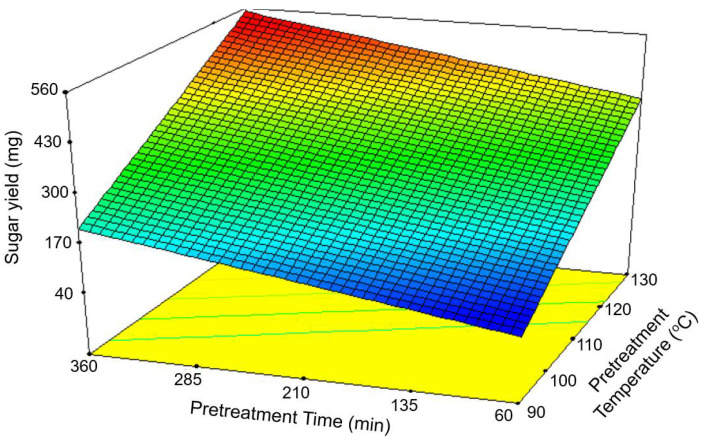
Response surface plot for the visualization of pretreatment effects (pretreatment temperature and pretreatment time) on sugar yield.

**Figure 2 bioengineering-09-00017-f002:**
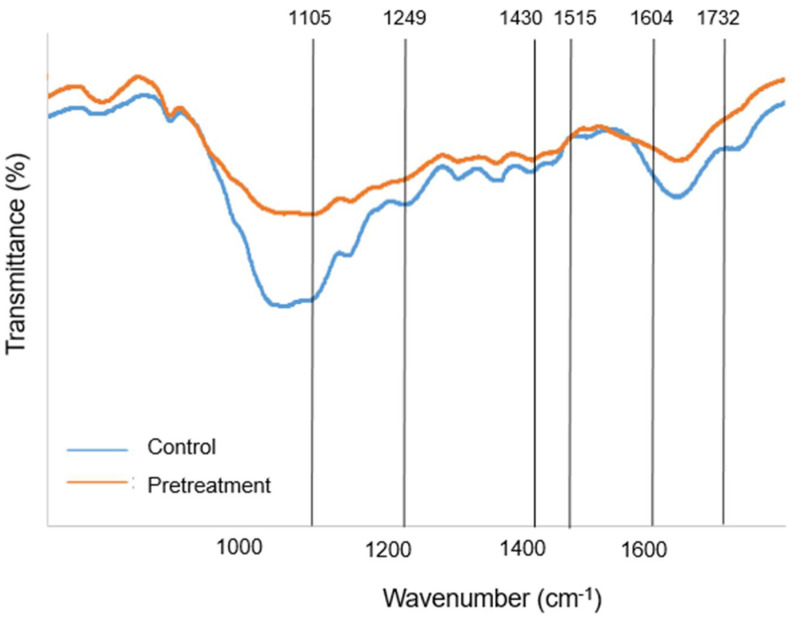
FTIR analysis of [Ch][OAc]-pretreated sample with selected wavenumbers representing changes in the functional groups of lignocellulose biomass.

**Figure 3 bioengineering-09-00017-f003:**
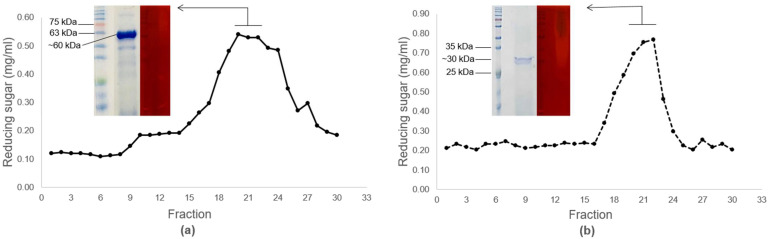
Evaluation of cellulase activities of (**a**) CBD2 and (**b**) CBD3, in the form of reducing sugars, in cellulase fractions obtained from size-exclusion liquid chromatography. The fractions with high cellulase activities were analyzed by SDS-PAGE analysis (blue color) and zymogram analysis (red color).

**Figure 4 bioengineering-09-00017-f004:**
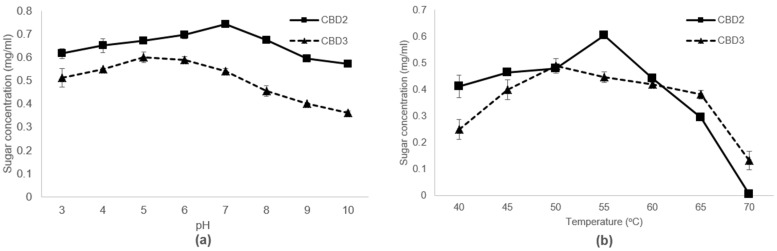
Optimal working conditions, (**a**) pH and (**b**) temperature, of bacterial cellulases produced from CBD2 and CBD3.

**Figure 5 bioengineering-09-00017-f005:**
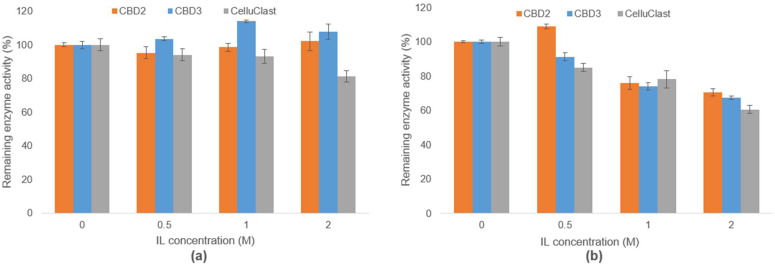
The remaining cellulase activities (CMCase) produced from CBD2 and CBD3 and commercial cellulase, Celluclast 1.5 L, under the influence of ILs, (**a**) [Ch][OAc] and (**b**) [EMIM][OAc].

**Figure 6 bioengineering-09-00017-f006:**
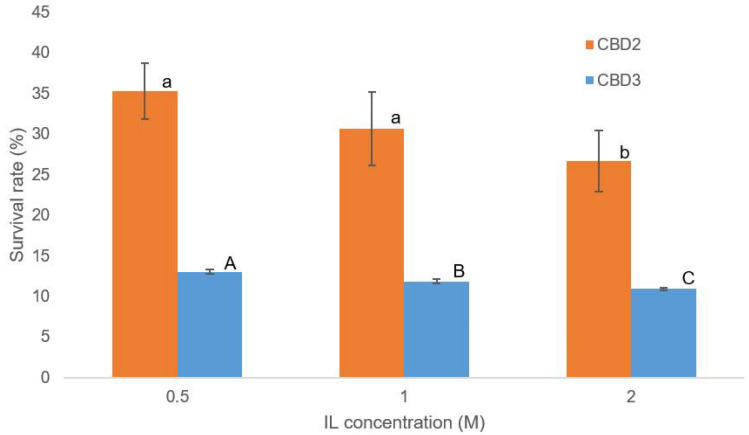
The survival rate of CBD2 and CBD3 cells under treatment of [Ch][OAc] at 0.5, 1.0 and 2.0 M. The represented data are the mean of three replicates ± standard error (*p* ≤ 0.05). ^a, b^ determine statistically difference in survival rate of CBD2. ^A, B, C^ determine statistically difference in survival rate of CBD3.

**Figure 7 bioengineering-09-00017-f007:**
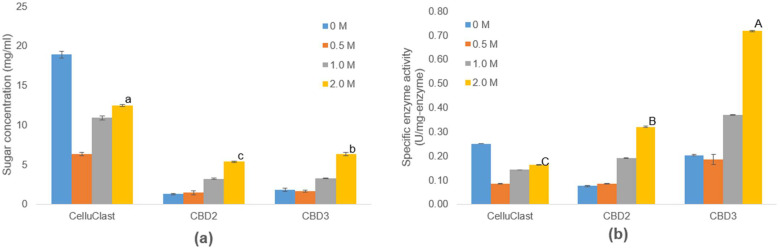
(**a**) The sugar concentrations produced from one-pot process and (**b**) specific enzyme activities of bacterial cellulase (CBD2 and CBD3) and commercial cellulase (Celluclast 1.5 L). The represented data are the mean of three replicates ±standard error (*p* ≤ 0.05). ^a, b, c^ determine statistically difference in sugar concentrations. ^A, B, C^ determine statistically difference in specific enzyme activity.

**Table 1 bioengineering-09-00017-t001:** [Ch][OAc] pretreatment conditions on rice straw based on RSM with Box-Behnken design. The reducing sugar yield (Y) was measured after pretreatment of each condition with different temperature (X_1_), time (X_2_), and loading ratio (X_3_).

Run	Pretreatment Temperature (°C)	Pretreatment Time (min)	Loading Ratio (%wt)	Sugar Yield (mg/g-Biomass)
1	90	60	10	120.47
2	90	210	5	145.46
3	90	210	15	138.06
4	90	360	10	217.98
5	110	60	5	159.76
6	110	60	15	176.04
7	110	210	10	287.54
8	110	210	10	294.45
9	110	210	10	340.33
10	110	210	10	335.89
11	110	210	10	291.49
12	110	360	5	362.04
13	110	360	15	408.91
14	130	60	10	431.60
15	130	210	5	512.02
16	130	210	15	496.23
17	130	360	10	528.30

**Table 2 bioengineering-09-00017-t002:** Analysis of variance (ANOVA) of the generated RSM model to represent the effect of [Ch][OAc] pretreatment on sugar yield.

Source	Sum of Squares	df	Mean Square	F-Value	*p*-Value (Prob > F)
Model	3.014 × 10^5^	2	1.507 × 10^5^	146.21	<0.0001
A-Temp	2.437 × 10^5^	1	2.437 × 10^5^	236.44	<0.0001
B-Time	57,690.11	1	57,690.11	55.98	<0.0001
Residual	14,427.98	14	1030.57		
Lack of Fit	11,748.81	10	1174.88	1.75	0.3095
Pure Error	2679.17	4	669.79		
Cor Total	3.158 × 10^5^	16			

**Table 3 bioengineering-09-00017-t003:** Validation of predicted RSM model by measurement of sugar yields and effect of pretreatment on lignocellulose compositions.

Sample	Sugar Yield (mg/g-Biomass)			Composition (%)		
	Predicted	Experimental	Error (%)	Cellulose	Hemicellulose	Lignin
Control	N.A.	90.5	N.A.	40.52 ± 0.32	32.70 ± 0.13	14.95 ± 0.20
Pretreat	542.3	551.2	1.64	45.84 ± 0.64	34.03 ± 0.17	8.39 ± 0.26
Predicted Model: Sugar yield (mg/g-biomass) = −773.09380 + 8.72626 × Time + 0.56613 × Time Optimal condition: Pretreatment temperature, 129.21 °C; Pretreatment time, 331.82 min; Loading ratio, 10.68%						

**Table 4 bioengineering-09-00017-t004:** CMCase activities and specific enzyme activities at the optimal working condition of crude cellulases produced from CBD2 and CBD3 and commercial Celluclast 1.5 L.

Enzyme	Optimal Condition	Enzyme Activity (U/mL)	Specific Enzyme Activity (U/mg)
CBD2	pH 7, 55 °C	0.62 ± 0.023 ^c^	0.20 ± 0.014 ^C^
CBD3	pH 5, 50 °C	0.73 ± 0.017 ^b^	0.22 ± 0.018 ^B^
Celluclast 1.5 L	pH 5, 50 °C	4.20 ± 0.033 ^a^	0.32 ± 0.025 ^A^

The represented data are the mean of three replicates ± standard error (*p* ≤ 0.05). ^a, b, c^ determine statistically difference in enzyme activities. ^A, B, C^ determine statistically difference in specific enzyme activities.

## Data Availability

The data of this study is included as reported here.

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
