# Peer review of "One-Pot Ionic Liquid-Mediated Bioprocess for Pretreatment and Enzymatic Hydrolysis of Rice Straw with Ionic Liquid-Tolerance Bacterial Cellulase"

_bioengineering, 2022, doi:10.3390/bioengineering9010017_

Round 1
Reviewer 1 Report
I reviewed the manuscript entitled, One-pot ionic Liquid-mediated Bioprocess for Pretreatment and Enzymatic Hydrolysis of Rice Straw with Ionic Liquid-tolerance Bacterial Cellulase. The introduction is well-written. Methodology should be improved with appropriate reference citation. Typos should be avoided and statistical analysis must be conducted. Figures quality should be improved.
Line 103: write the model no and machine suppliers city and country.
Line 105: which sieve, US?
Lines 109 and 143: scientific name should be in Italics
Line 135: centrifuged; write the model no and machine suppliers city and country.
Line 145: water bath, write the model no and machine suppliers city and country.
Line 162 and 176: NaNO3.. soluble be revised with proper notation
Line 184: Sigmaaldrich… revise to Sigma Aldrich
Section 2.5. Please provide details of SDS-PAGE methodology
Section 2.6.: give citation for this method
Section 2.7. give citation for this method
Many methodologies are without references. Please revise and cite references
All figures must be improved for its resolution and statistical analysis must be performed, wherever applicable, to understand the differences.
For all figures, Y axis should be provided
References are not according to the journal format. Please revise accordingly
Table 1. convert E+005 values to scientific such as 10^5 etc
Please represent choline acetate either with ([Ch][OAc]) or [Ch][Oac]. Fix to one way of representation
Figure 3. standard marker SDS-PAGE should be provided for comparison purposes.
Table 4. statistical analysis must be performed
Author Response
Reviewer comment: I reviewed the manuscript entitled, One-pot ionic Liquid-mediated Bioprocess for Pretreatment and Enzymatic Hydrolysis of Rice Straw with Ionic Liquid-tolerance Bacterial Cellulase. The introduction is well-written. Methodology should be improved with appropriate reference citation. Typos should be avoided and statistical analysis must be conducted. Figures quality should be improved.
Authors response: Thank you for reviewer comments and opportunity to revise this manuscript. The methodology of this submitted manuscript was revised by adding appropriate reference citation as suggested. Typos were checked for the whole manuscript and statistical analysis was conducted. The figure quality was enhanced.
Reviewer comment: Line 103: write the model no and machine suppliers city and country.
Authors response: The details of model no and machine suppliers city and country were added as suggested (Line 103-104).
Reviewer comment: Line 105: which sieve, US?
Authors response: It is 20 mesh based on US mesh chart (reference: https://www.azom.com/article.aspx?ArticleID=1417)
Reviewer comment: Lines 109 and 143: scientific name should be in Italics
Authors response: The scientific names were changed to Italics.
Reviewer comment: Line 135: centrifuged; write the model no and machine suppliers city and country.
Authors response: The details of model no and machine suppliers city and country were added as suggested (Line 136).
Reviewer comment: Line 145: water bath, write the model no and machine suppliers city and country.
Authors response: The details of model no and machine suppliers city and country were added as suggested (Line 147).
Reviewer comment: Line 162 and 176: NaNO3.. soluble be revised with proper notation
Authors response: The notation of NaNO3 was corrected
Reviewer comment: Line 184: Sigmaaldrich… revise to Sigma Aldrich
Authors response: it was revised to Sigma Aldrich
Reviewer comment: Section 2.5. Please provide details of SDS-PAGE methodology
Authors response: The detail of SDS-PAGE methodology was added
Reviewer comment: Section 2.6.: give citation for this method
Authors response: The references of this section were added.
Reviewer comment: Section 2.7. give citation for this method
Authors response: The reference of this section was added.
Reviewer comment: Many methodologies are without references. Please revise and cite references
Authors response: Authors checked and added references as suggested.
Reviewer comment: All figures must be improved for its resolution and statistical analysis must be performed, wherever applicable, to understand the differences.
Authors response: Authors conducted the statistical analysis as suggested for results in Table 4, Figure 6 and 7
Reviewer comment: For all figures, Y axis should be provided
Authors response: All figures have Y-axis labels and names
Reviewer comment: References are not according to the journal format. Please revise accordingly
Authors response: The formats of all references were revised.
Reviewer comment: Table 1. convert E+005 values to scientific such as 10^5 etc
Authors response: the values were converted to scientific format
Reviewer comment: Please represent choline acetate either with ([Ch][OAc]) or [Ch][Oac]. Fix to one way of representation
Authors response: All writing was revised to [Ch][OAc]
Reviewer comment: Figure 3. standard marker SDS-PAGE should be provided for comparison purposes.
Authors response: the standard marker for SDS-PAGE and zymogram was added
Reviewer comment: Table 4. statistical analysis must be performed
Authors response: The statistical analysis of Table 4 was conducted
Reviewer 2 Report
Bioengineering-MDPI
Manuscript number: bioengineering-1540543
Title: One-pot ionic Liquid-mediated Bioprocess for Pretreatment and Enzymatic Hydrolysis of Rice Straw with Ionic Liquid-tolerance Bacterial Cellulase
The manuscript “One-pot ionic Liquid-mediated Bioprocess for Pretreatment and Enzymatic Hydrolysis of Rice Straw with Ionic Liquid-tolerance Bacterial Cellulase“ is a paper that reports statistical optimization of IL pretreatment and enzymatic saccharification of rice straw using low toxic choline acetate and IL tolerant bacterial cellulase.
The idea of using one pot process is very interesting and deserves attention. The main criticism is reflected to the model, Box-Benken design under the RSM.
Page 3, 2.2 Lignocellulose pretreatment and enzymatic saccharification -The authors used Box-Benken design for the optimization process.
The Box-Benken design was used to develop the non-linear models- quadratic models, please see the reference (https://www.statease.com/docs/v11/designs/rsm/#rsm.)
In this study, the authors used Box-Benken design to navigate their experiments (17 experiments), but the model was not quadratic but linear! (page 7, line 15)- “The Linear vs. Mean model was suggested to represent the experimental data of [Ch][Oac] pre….”- the Design Expert suggested that model was linear…it does not mean that you must use this linear model….In nature, the biological processes are not linear and because of this scientists are using mathematical models that are not linear…
In this paper, increasing the pretreatment time and pretreatment temperature the sugar yield is also increasing, but up to what value? Is there some boundary, some up level, for this? If is this a linear model, there is no limit value.
I suggest authors to do their experiments again, to carefully read the manual about Box-Benken design, and to present the model in a quadratic form. In that way, their model will be reliable.
Author Response
Reviewer comment:
The manuscript “One-pot ionic Liquid-mediated Bioprocess for Pretreatment and Enzymatic Hydrolysis of Rice Straw with Ionic Liquid-tolerance Bacterial Cellulase“ is a paper that reports statistical optimization of IL pretreatment and enzymatic saccharification of rice straw using low toxic choline acetate and IL tolerant bacterial cellulase.
The idea of using one pot process is very interesting and deserves attention. The main criticism is reflected to the model, Box-Benken design under the RSM.
Page 3, 2.2 Lignocellulose pretreatment and enzymatic saccharification -The authors used Box-Benken design for the optimization process.
The Box-Benken design was used to develop the non-linear models- quadratic models, please see the reference (https://www.statease.com/docs/v11/designs/rsm/#rsm.)
In this study, the authors used Box-Benken design to navigate their experiments (17 experiments), but the model was not quadratic but linear! (page 7, line 15)- “The Linear vs. Mean model was suggested to represent the experimental data of [Ch][Oac] pre….”- the Design Expert suggested that model was linear…it does not mean that you must use this linear model….In nature, the biological processes are not linear and because of this scientists are using mathematical models that are not linear…
In this paper, increasing the pretreatment time and pretreatment temperature the sugar yield is also increasing, but up to what value? Is there some boundary, some up level, for this? If is this a linear model, there is no limit value.
I suggest authors to do their experiments again, to carefully read the manual about Box-Benken design, and to present the model in a quadratic form. In that way, their model will be reliable.
Authors response: Thank you the reviewer’s comments for constructive suggestion on the RSM results. We agreed to reviewer that the RSM with BBD normally preferred the quadratic model. The quadratic model could help researchers to find the “optimal point” within the “tested boundary”. In our study, under tested boundaries of three factors, and among all models, linear model was suggested with the highest P-value (< 0.0001) and R2 value of 0.9528. These analysis of statistic fit is reliable scientifically. We also agreed as reviewer that the perfect model obtained from RSM should be the quadratic model, and we previously applied RSM with BB design with other pretreatment studies. For example,
- Amnuaycheewa P, Hengaroonprasan R, Rattanaporn K, Kirdponpattara S, Cheenkachorn K, Sriariyanun M. 2016. Enhancing enzymatic hydrolysis and biogas production from ricestraw by pretreatment with organic acids. Industrial Crops and Products. 84:247-254.
- Rattanaporn K, Tantayotai P, Phusantisampan T, Pornwongthong P, Sriariyanun M. 2018. Organic acid pretreatment of oil palm trunk: effect on enzymatic saccharification and ethanol production. Bioprocess and Biosystem Engineering. 41:467-477.
- Sriariyanun M, Yan Q, Nowik I, Cheenkachorn K, Phusantisampan T, Modigell M. 2015. Efficient pretreatment of rice straw by combination of screw press and ionic liquid to enhance enzymatic hydrolysis. Kasetsart Journal (Natural Science) vol 49, no. 1. p 146-154.
Among these works, different chemical pretreatments, including imidazolium-derived ionic liquid pretreatment, were optimized by RSM with BBD. Using the same RSM methodology, same design, same pretreatment factor, just changing tested boundaries, we obtained different models from these experiments, including quadratic and 2FI. Since RSM model explain the relationships to all independent factors, there are possibilities that some factors insignificant affect dependent factor, or response factor. The RSM results in this submitted manuscript showed that the best fit model to represent our experimental data is linear model and supported by p-value and R2. Although, authors agreed with reviewer that both pretreatment time and temperature should be interacting factors and their interactions should positively or negatively synergy in a “certain range” of “tested boundary”. Additionally, in our RSM design, the tested boundaries of pretreatment parameters were selected based on literature studies or ionic liquid pretreatment in our previous published works and other related works published by other labs. For example;
- Sriariyanun M, Yan Q, Nowik I, Cheenkachorn K, Phusantisampan T, Modigell M. 2015. Efficient pretreatment of rice straw by combination of screw press and ionic liquid to enhance enzymatic hydrolysis. Kasetsart Journal (Natural Science) vol 49, no. 1. p 146-154.
- https://pubs.rsc.org/en/content/articlelanding/2014/gc/c3gc42401d
- https://www.sciencedirect.com/science/article/pii/S0960852415007373?via%3Dihub
- https://www.sciencedirect.com/science/article/abs/pii/S1385894717318284?via%3Dihub
- https://www.sciencedirect.com/science/article/pii/S096085241401640X?via%3Dihub
Additionally, although we cannot get the quadratic model in this work, however, we get the maximum yields of sugars within the ranges of our tested boundary. This is the target of our work. After we obtained the maximum sugar yields in our scope of study, we therefore used this pretreatment condition to combine with enzymatic saccharification and conducted one-pot process. Lastly, we really appreciated reviewer suggestion to repeat experiments again, although these results were already repeated at least 3 times already. We would like to ask for your understanding that we cannot repeat this experiment in short time period, due to covid19 situation, and the shut-down of laboratory facility. Also, this pretreated sample was not only used for preparation of maximum sugar yield, but it was also used as raw materials in one-pot process. To repeat this one-pot reaction, in addition to repeat the RSM, we also need to repeat cellulase preparation and enzyme activity analysis. So, it is difficult to repeat these experiments in short time periods.
Round 2
Reviewer 1 Report
Authors thoroughly answered the questions raised by me. In my opinion, the present version of the manuscript can be accepted for publication.
Reviewer 2 Report
In this form, this paper could be published in Bioengineering.